# Duck Plague Virus Promotes DEF Cell Apoptosis by Activating Caspases, Increasing Intracellular ROS Levels and Inducing Cell Cycle S-Phase Arrest

**DOI:** 10.3390/v11020196

**Published:** 2019-02-24

**Authors:** Chuankuo Zhao, Mingshu Wang, Anchun Cheng, Qiao Yang, Ying Wu, Renyong Jia, Dekang Zhu, Shun Chen, Mafeng Liu, Xinxin Zhao, Shaqiu Zhang, Yunya Liu, Yanling Yu, Ling Zhang, Bin Tian, Mujeeb Ur Rehman, Leichang Pan, Xiaoyue Chen

**Affiliations:** 1Institute of Preventive Veterinary Medicine, Sichuan Agricultural University, Wenjiang, Chengdu 611130, Sichuan, China; zck0523@163.com (C.Z.); mshwang@163.com (M.W.); yangqiao721521@sina.com (Q.Y.); yingzi_no1@126.com (Y.W.); jiary@sicau.edu.cn (R.J.); shunchen@sicau.edu.cn (S.C.); liumafengra@163.com (M.L.); xxinzhao@163.com (X.Z.); shaqiu86@hotmail.com (S.Z.); yunnyaaliu@163.com (Y.L.); yanling3525@163.com (Y.Y.); zl97451@126.com (L.Z.); btian_1985@163.com (B.T.); mujeebnasar@yahoo.com (M.U.R.); pl2007@126.com (L.P.); 2Key Laboratory of Animal Disease and Human Health of Sichuan Province, Sichuan Agricultural University, Wenjiang, Chengdu 611130, Sichuan, China; zdk24@sicau.edu.cn (D.Z.); chenxy_24@sina.cn (X.C.); 3Avian Disease Research Center, College of Veterinary Medicine, Sichuan Agricultural University, Wenjiang, Chengdu 611130, Sichuan, China

**Keywords:** apoptosis, cell cycle, duck embryo fibroblast, duck plague virus, ROS

## Abstract

Background: Duck plague virus (DPV) can induce apoptosis in duck embryo fibroblasts (DEFs) and in infected ducks, but the molecular mechanism of DPV-induced apoptosis remains unknown. Methods: We first used qRT-PCR and a Caspase-Glo assay to determine whether the caspase protein family plays an important role in DPV-induced apoptosis. Then, we used an intracellular ROS detection kit and the mitochondrial probe JC-1 to respectively detect ROS levels and mitochondrial membrane potential (MMP). Finally, flow cytometry was used to detect apoptosis and cell cycle progression. Results: In this study, the mRNA levels and enzymatic activities of caspase-3, caspase-7, caspase-8, and caspase-9 were significantly increased during DPV-induced apoptosis. The caspase inhibitors Z-DEVD-FMK, Z-LEHD-FMK, and Q-VD-Oph could inhibit DPV-induced apoptosis and promote viral replication. Subsequently, a significant decrease in MMP and an increase in the intracellular ROS levels were observed. Further study showed that pretreating infected cells with NAC (a ROS scavenger) decreased the intracellular ROS levels, increased the MMP, inhibited apoptosis, and promoted viral replication. Finally, we showed that DPV infection can cause cell cycle S-phase arrest. Conclusions: This study shows that DPV causes cell cycle S-phase arrest and leads to apoptosis through caspase activation and increased intracellular ROS levels. These findings may be useful for gaining an understanding of the pathogenesis of DPV and the apoptotic pathways induced by α-herpesviruses.

## 1. Introduction

Duck plague is an acute, contagious, and fatal disease in waterfowl (ducks and geese) caused by duck plague virus (DPV), a member of the α-herpesvirus subfamily with a double-stranded DNA genome of approximately 162 kb, and a capsid, tegument, and envelope [1,2,3,4,5,6]. Virulent DPV strains are highly pathogenic and induce massive petechial hemorrhages in parenchymal organs, lymphoid tissue, and the digestive tract; they also cause large numbers of duck deaths [7,8,9,10]. To control this disease on duck farms, attenuated DPV vaccines and DNA vaccines have been studied [11,12,13,14,15,16].

Apoptosis is an indispensable innate immune mechanism that can effectively eliminate infected cells [17]. The morphological characteristics of apoptosis include chromatin aggregation and condensation, and apoptotic body formation. Apoptosis has been shown to be induced by two classical pathways: the intrinsic and extrinsic pathways [18]. The intrinsic signaling pathway is known as mitochondria- and endoplasmic reticulum (ER)-initiated apoptosis. Mitochondria are fundamentally involved in the mitochondrial apoptosis pathway, which is regulated by numerous factors, such as members of the Bcl-2 protein family. Bcl-2 protein family members, including Bax, Bak, Bcl-2, Bcl-xl, Mcl-1, Bid, and Bim, decrease the mitochondrial membrane potential (MMP), causing the release of Cyt-c, which forms a complex with pro-caspase-9 and Apaf-1. This complex activates caspase-9, which then activates downstream caspase-3 to initiate apoptosis [19]. In cells, ER stress regulates the concentration of Ca^2+^ and initiates the IRE1, PERK, and ATF6 pathways, which are associated with the mitochondrial pathway of apoptosis [20,21]. Additionally, increased ROS levels accelerate apoptosis. ROS is produced by the mitochondria, which are affected by ROS levels. Increased ROS levels reduce the MMP, which induces the mitochondrial apoptosis pathway [22,23,24]. By contrast, the extrinsic pathway is activated by the death ligand and its corresponding receptor. Subsequently, pro-caspase-8 is recruited to form the death-inducing signaling complex (DISC), leading to the activation of caspase-8, which activates downstream caspase-3 to induce apoptosis [25,26].

The results of many studies have revealed that the molecular mechanism of α-herpesvirus-induced apoptosis is complicated. Moreover, previous findings have proven that α-herpesviruses can encode many proteins that inhibit apoptosis, including Us3, Us5, ICP4, ICP22, ICP27, and LAT [18]. α-Herpesviruses can modulate apoptosis, which plays an important role in viral replication and latent infection. The results of our previous studies have demonstrated that DPV causes apoptosis in infected ducks and duck embryo fibroblasts (DEFs) [27,28], but the molecular mechanism of this activity remains unknown. The results of this study show that DPV causes cell cycle S-phase arrest and leads to apoptosis through caspase activation and increased intracellular ROS levels, providing a basis for further studies on DPV pathogenesis and the apoptotic pathways induced by α-herpesviruses.

## 2. Materials and Methods

### 2.1. Cells and Viruses

DEFs were prepared from 9- to 11-day-old duck embryos (The farm of Sichuan Agricultural Uniersity). The cells were grown in Eagle’s minimal essential medium (MEM) (Sigma, St. Louis, MO, USA) containing 10% newborn bovine serum (NBS) (Gibco, Gaithersburg, MD, USA) at 37 °C in a humidified incubator with 5% CO_2_. The DPV CHv strain was previously isolated and characterized in our laboratory [29,30]. Unless otherwise stated, the median tissue culture infective dose (TCID_50_) used in DEFs was 1 MOI. DEFs were infected with DPV CHv for 1 h at 37 °C, and were then cultured in MEM supplemented with 2% NBS.

### 2.2. Cell Viability

Cell viability was measured using an MTT assay kit (Sangon Biotech, Shanghai, China) according to the manufacturer’s instructions. Briefly, the cells were seeded in 96-well plates. After pretreatment with Z-DEVD-FMK, Z-IETD-FMK, Z-LEHD-FMK, Q-VD-Oph, or NAC for 2 h, the cells were incubated in culture medium containing MTT (0.5 mg mL^−1^) for 4 h at 37 °C. The medium was subsequently replaced with a formazan solubilization solution, and the absorbance at 570 nm was measured using a microplate reader.

### 2.3. Detection of DPV

Viral DNA was extracted using a MiniBEST Viral RNA/DNA extraction kit (TaKaRa, Dalian, China) and then used for quantitative real-time PCR; this method is described in detail in our previously published article [31]. The following primers were generated based on the viral gene UL30: Forward primer—5′-GGACAGCGTACCACAGATAA-3′, Reverse primer—5′-ACAAATCCCAAGCGTAG-3′. DPV was detected by determining the TCID_50_ [32]. Briefly, serial 10-fold dilutions of the viral supernatant were made, and 100 μL of each dilution was inoculated into eight wells of a 96-well microtitration plate with an appropriate cell culture monolayer. The plates were incubated for 5–7 days at 37 °C in a 5% CO_2_ atmosphere, after which the cultures were examined for cytopathology under a light microscope. The dilution of the suspension that caused cytopathology in half of the cultures (TCID_50_/0.1 mL) was calculated according to the method of Spearman and Kaerber.

### 2.4. DAPI Staining of Cell Nuclei

DPV-infected and control cells were collected and fixed with 4% paraformaldehyde for 60 min. Subsequently, the solution was permeabilized for 30 min and stained with DAPI for 10 min. The nuclei were observed with a fluorescence microscope.

### 2.5. Isolation of RNA and Analysis of the mRNA Expression of Apoptosis-Related Genes by qRT-PCR

At different time points postinfection, total RNA was extracted from DPV-infected cells and mock-infected cells using RNAiso Plus (TaKaRa, Dalian, China) according to the manufacturer’s instructions. The extraction yield was assessed by measuring the absorbance at 260 nm, and RNA quality was evaluated by calculating the ratio of the absorbance at 260 nm to that at 280 nm. Extracted RNA was immediately reverse-transcribed into first-strand cDNA using a PrimeScript RT Reagent kit with gDNA Eraser. For qRT-PCR, a 20 µL total reaction volume containing 10 µL of SYBR Premix Ex Taq II (Tli RNaseH Plus), 1 µL of forward primer, 1 µL of reverse primer, 6 µL of RNase-free water, and 2 µL of cDNA was used. The thermal cycling procedure included initial denaturation for 30 s at 95 °C, followed by 40 cycles of 5 s at 95 °C and 30 s at the melting temperature of the specific primer pair. Gene expression was analyzed, and β-actin was used as the internal control. The accession numbers listed in Table 1 were obtained from NCBI GenBank. The relative gene expression was calculated using the mean values obtained with the arithmetic formula ΔΔCt.

### 2.6. Flow Cytometric Analysis of Apoptosis

Apoptotic cells were detected by flow cytometry (FCM). First, 5 µL of Via-Probe™ (cat. no. 555816; BD Pharmingen, USA) and 5 µL of Annexin-PE (BD Pharmingen) were added to 100 µL of cell suspension and were incubated at 25 °C in the dark for 15 min. Annexin binding buffer (450 µL) (BD Pharmingen, USA, 51-66121E) was then added to the mixture, and the percentage of apoptotic cells was assayed by FCM within 1 h.

### 2.7. FCM Cell Cycle Analysis

Cell cycle analysis was performed by FCM using DPV-infected and control cells that were collected and fixed using 70% ethanol. The cells were subsequently washed with PBS twice and then incubated with 0.5 mL of a PI/RNASE solution for 15 min. The cells were detected by FCM within 1 h.

### 2.8. Determination of Caspase-3, Caspase-7, Caspase-8, and Caspase-9 Activities

The activities of caspase-3, caspase-7, caspase-8, and caspase-9 were measured using a Caspase-Glo assay kit (Promega, Madison, Wisconsin, USA) according to the manufacturer’s instructions. The cells were seeded into 12-well plates and were counted with a blood cell counting plate. From each sample, approximately 20,000 cells were added to 100 µL of Caspase-Glo reagent in a 96-well white-walled plate and incubated for 30 min. Luciferase activity was detected using a multifunctional microplate reader (Thermo Scientific, Massachusetts, USA), and the fold increase in protease activity was determined by comparing the luciferase activity of the infected cells with that of mock-infected cells. In the caspase inhibitor assays, caspase inhibitors were dissolved in dimethyl sulfoxide (DMSO) and stored at −20 °C before use. Cells were exposed to caspase inhibitors (i.e., 20 mM Z-DEVD-FMK, Z-IETD-FMK, Z-LEHD-FMK, and Q-VD-Oph) for 2 h prior to DPV infection.

### 2.9. Determination of MMP

MMP was measured using a JC-1 staining kit (Sigma-Aldrich, Shanghai, China) according to the manufacturer’s instructions. Briefly, MEM-washed cells were stained with JC-1 dye for 20 min. Then, the cells were observed, and the data were collected with a fluorescence microscope and a multifunctional microplate reader.

### 2.10. Determination of Intracellular ROS Levels

The intracellular ROS levels of cells were measured using an intracellular ROS detection kit (Sigma-Aldrich) according to the manufacturer’s instructions. Briefly, the cells were inoculated into a 96-well microtitration plate, 100 µL of ROS detection reagent mixture was added, and the plate was incubated for 1 h at 37 °C in a 5% CO_2_ humidified incubator. Subsequently, the data were collected using a fluorescence microscope and a multifunctional microplate reader.

### 2.11. Statistical Analysis

The data were assessed using GraphPad Prism 6, and statistical significance was assessed using Student’s *t*-test. The data are presented as the means ± SD; * *p* < 0.05 and ** *p* < 0.01 indicate significance compared with the control.

## 3. Results

### 3.1. Cytopathic Effects (CPEs) Induced by DPV in DEFs

First, the morphological changes in DPV-infected DEFs were determined by microscopic observations 12, 24, 36, 48, and 60 h postinfection (hpi) (Figure 1A). At 36, 48, and 60 hpi, compared with the morphology of the control cells, obvious cellular fragmentation and plaques were observed in the DPV-infected DEFs. The arrows indicate that the infected cells appeared with CPEs at 24, 36, 48, and 60 hpi. 4’,6-Diamidino-2-phenylindole (DAPI) staining was performed to observe the morphological changes of the cell nuclei (Figure 1B), and syncytia were present at 36 and 48 hpi in the DPV-infected cells, which is denoted by arrowheads. The above observations showed that DPV causes CPEs in DEFs. In addition, DAPI staining at 24, 36, 48, and 60 hpi revealed the presence of apoptosis-associated morphological changes, such as nuclear fragmentation and apoptotic bodies. At 24, 36, 48, and 60 hpi, the arrows indicate that the nuclei of infected cells appear as marginated typical apoptotic bodies. We used quantitative real-time PCR [31] and median tissue culture infective dose (TCID_50_) assays to detect DPV (Figure 1C,D); the results show that the viral DNA and titers gradually increased as the infection progressed.

### 3.2. Effect of DPV Infection on Caspases

Next, we determined whether the caspase protein family plays an important role in DPV-induced apoptosis. The mRNA levels of caspase-3, caspase-7, caspase-8, and caspase-9 were detected by qRT-PCR. As shown in Figure 2A, compared with control cells, DPV-infected cells exhibited significant increases in caspase-3 and caspase-9 mRNA levels at 12, 24, 36, 48, and 60 hpi, while the caspase-7 mRNA level was significantly increased at 12, 24, 36, and 48 hpi in the infected cells. Compared with the control cells, the infected cells exhibited significant increases in the caspase-8 mRNA level at 24, 36, 48, and 60 hpi. The results in Figure 2B show that caspase-8 activity was significantly higher in infected cells than in control cells at 48 and 60 hpi, and caspase-3 and caspase-7 activities were significantly higher in infected cells than in control cells at 12, 24, 36, 48, and 60 hpi. Furthermore, caspase-9 activity was significantly higher in infected cells than in control cells at 24, 36, 48, and 60 hpi.

### 3.3. Caspase Inhibitors Inhibit DPV-Induced Apoptosis and Promote Viral Replication

To further assess the role of the caspase protein family in apoptosis, the effects of Z-DEVD-FMK, Z-IETD-FMK, and Z-LEHD-FMK (specific inhibitors of caspase-3, caspase-8, and caspase-9, respectively), and the pan-caspase inhibitor Q-VD-Oph were determined. Cells that were pretreated with Z-DEVD-FMK, Z-IETD-FMK, Z-LEHD-FMK, and Q-VD-Oph for 2 h and detected at 48 h showed no signs of cytotoxicity (Figure 3A,B). Cells were also pretreated with Z-DEVD-FMK, Z-IETD-FMK, Z-LEHD-FMK, and Q-VD-Oph for 2 h and infected with DPV. At 36 hpi, Z-DEVD-FMK and Z-LEHD-FMK inhibited DPV-induced apoptosis (Figure 3D,E), while at 48 hpi, Q-VD-Oph but not Z-IETD-FMK inhibited DPV-induced apoptosis (Figure 3D,E). Finally, we assessed the effect of Z-DEVD-FMK, Z-LEHD-FMK, Z-IETD-FMK, and Q-VD-Oph on viral replication. The results in Figure 3C show that Z-DEVD-FMK and Q-VD-Oph effectively promoted viral replication, whereas Z-LEHD-FMK and Z-IETD-FMK had a weak effect on viral replication.

### 3.4. Effect of DPV Infection on ROS

To determine which apoptotic pathway is induced by DPV, an intracellular ROS detection kit was used to assess ROS production in DPV-infected cells. The results presented in Figure 6A show that at 2, 4, 12, 24 and 36 hpi, red fluorescence was stronger in the DPV-infected cells than in the control cells, while this fluorescence began to weaken at 48 hpi (Figure 4A). The red fluorescence value observed for DPV-infected cells was significantly higher than that of the control cells at 2, 4, 12, 24, and 36 hpi but was not significantly different from that of the control cells at 48 hpi (Figure 4B).

### 3.5. Effect of DPV Infection on MMP

ROS are produced by the mitochondria, which are affected by ROS levels. Increased ROS levels reduce the MMP, which induces the mitochondrial apoptosis pathway. To assess the changes in the MMP induced by DPV, the mitochondrial JC-1 probe was used. In normal cells, where the MMP is high, JC-1 aggregates in the mitochondria, forming polymers and emitting red fluorescence. By contrast, in apoptotic cells, where the MMP is reduced, JC-1 cannot aggregate in the mitochondria and remains in the monomeric form, emitting green fluorescence. The fluorescence microscopy results showed that compared with the control cells, the green fluorescence in DPV-infected DEFs was significantly increased (Figure 5A). The JC-1 aggregate to monomer ratio was significantly lower in the DPV-infected DEFs than in the control cells (Figure 5B). From these results, we concluded that DPV induces apoptosis in DEFs, leading to a decrease in MMP.

### 3.6. NAC Scavenges Intracellular ROS, Increases the MMP, Inhibits Apoptosis and Promotes Viral Replication

To further assess the role of ROS in the apoptotic process, we evaluated the MMP and apoptosis in cells treated with NAC. The cells pretreated with 5 or 10 mM NAC for 2 h and detected at 36 h showed no signs of cytotoxicity (Figure 6A), and in the DPV-infected cells, NAC scavenged ROS (Figure 6B), increased the MMP (Figure 6C), and inhibited apoptosis (Figure 6E). Apoptosis is an immune defense mechanism of host cells, and because NAC can inhibit the apoptosis induced by DPV, treating cells with NAC can promote viral replication (Figure 6D).

### 3.7. DPV Induces DEF S-Phase Cell Cycle Arrest

FCM was used to assess the effect of DPV on cell cycle distribution. At 24 and 36 hpi, DPV-infected DEFs showed a significant decrease in the percentage of DEFs in the G0/G1-phase compared with the control cells, while the percentage of DEFs in the S-phase was significantly increased. In addition, the percentage of DEFs in the G2/M-phase was significantly decreased at 36 hpi, but this percentage was not different from that of the control cells at 24 hpi (Figure 7). Our results suggest that DPV causes S-phase arrest.

## 4. Discussion

Many studies have shown that α-herpesviruses can induce cell-specific apoptosis. In herpes simplex virus (HSV)-1-infected brain and skin cells, the virus can cause apoptosis in skin cells but not in brain cells [33]. Both lymphocytes and bovine kidney epithelial (MDBK) cells undergo apoptosis after being infected with BHV-1, but apoptosis does not occur in BHV-1-infected trigeminal nerves [34]. Our laboratory has discovered that DPV can induce apoptosis in the thymic, splenic, and pancreatic lymphocytes of adult ducks, and that it can cause apoptosis in DEFs in vitro [27,28]. However, there are large gaps in knowledge regarding the molecular mechanism of DPV-induced apoptosis. Although the molecular mechanism of α-herpesvirus-induced apoptosis has been studied, the understanding of this mechanism is incomplete. Previous studies have shown that α-herpesviruses induce apoptosis through the death receptor, and mitochondrial, ROS, P53, MAPK, and TLR signaling pathways [35,36,37,38]. The results of this study showed, for the first time, that DPV can induce cell cycle S-phase arrest and apoptosis through caspase activation and increased intracellular ROS levels. Further study of the apoptosis pathway induced by α-herpesviruses will provide additional helpful information.

The caspase protein family plays a major role in the process of cell apoptosis, with caspase-9 and caspase-8 being involved in the mitochondrial and death receptor pathways, respectively, which activate downstream caspase-3 to induce apoptosis. In this study, DPV was observed to activate caspase-3, caspase-7, caspase-8, and caspase-9, as evidenced by their increased mRNA levels and activities. For other α-herpesviruses, HSV-1 infection in mononuclear cells activates caspase-9 and induces apoptosis. Xu et al. [39] studied the molecular mechanism of BHV-1-induced apoptosis and observed that Cyt-c was released into the cytoplasm from the mitochondria to activate caspase-9-induced apoptosis. In this study, we assessed the effects of the caspase-3-, caspase-8-, and caspase-9- specific inhibitors Z-DEVD-FMK, Z-IETD-FMK, and Z-LEHD-FMK, and the pan-caspase inhibitor Q-VD-Oph, on DPV-infected cells. The results showed that cells treated with the specific inhibitors for caspase-3 and caspase-9 and the pan-caspase inhibitor had impaired DPV-induced apoptosis, whereas the caspase-8-specific inhibitor did not induce these effects. These results suggest that caspase-3, caspase-7, and caspase-9 play a substantial role in DPV-induced apoptosis but that caspase-8 has a lesser role in this process. Xu et al. [39] showed that in BHV-1-infected MDBK cells, the protein levels of Fas and FasL increased over time and in a BHV-1 dose-dependent manner, as did the activity of caspase-8, which is consistent with our results. Although our results showed that caspase-8 played a small role in DPV-induced apoptosis, we speculated that DPV-induced apoptosis may be facilitated through the death receptor pathway.

DPV, BHV-1, HSV-1, and HSV-2 all belong to the α-herpesviruses subfamily and share a number of characteristics. The interplay between ROS and BHV-1, HSV-1, or HSV-2 in various cell cultures has been extensively studied [40,41,42]. Based on the results of previous studies, we inferred that cellular ROS are broadly regulated by α-herpesviruses in a cell- and virus-specific manner, e.g., HSV-1 infection of murine microglial cells and neural cells increased ROS levels at 24–72 and 1–24 hpi, respectively [40,42]. The HSV-2-induced production of ROS in RAW246.7 cells could be notably detected at 1 hpi [43]. In addition, increased ROS levels in DEFs infected with BHV-1 were previously detected at 1–12 hpi [44]. In this study, increased ROS levels in DEFs infected with DPV were detected at 2–24 hpi. HSV infection has been shown to result in the loss of MMP and to decrease levels of cellular ATP at the late stage of infection [45]. BHV-1 infection in MDBK cells caused mitochondria dysfunction as demonstrated by mitochondrial depolarization and reduced ATP levels during the late stage of infection [44], although the role of ROS in these adverse effects has yet to be elucidated. Since mitochondria generate ROS and because increased levels of ROS can lower the MMP, it is unsurprising that mitochondria are a potential target of some viruses that are able to induce excessive ROS. The results of our study showed that DPV infects DEFs, leading to a decrease in MMP. Importantly, by treating cells with NAC, a ROS scavenger, we observed that DPV promotes ROS production in the early stage of infection, inhibiting the decrease in MMP. We also showed that NAC can inhibit apoptosis induced by DPV. However, the HSV-1 ICP27 protein is a multifunctional protein that induces apoptosis through the mitochondrial pathway. In addition, ICP27 also increases intracellular ROS levels and participates in mitochondrial pathway-induced apoptosis [46]. As noted previously, BHV-1-infected cells also exhibit increased ROS levels during the early stage of infection, and NAC can inhibit the reduction in MMP caused by BHV-1, which is consistent with the DPV results [39,44].

In this study, we observed a reduction in G0/G1-phase cells and an increase in S-phase cells 24 h after DEFs were infected with DPV, while fewer G2/M-phase cells were observed at 36 hpi, showing that DPV can cause S-phase arrest. Previous studies demonstrated that an α-herpesvirus infection can result in cell cycle arrest. By treating serum-starved cells with a mutated HSV strain that expresses only ICP0, it was previously shown that cells fail to enter into the S-phase after the addition of serum. Experiments synchronizing Hep2 cells transfected with ICP0-encoding plasmids showed that prior to transfection, these cells appeared to be arrested at the G1-phase of the cell cycle. However, following transfection with ICP0, the cells synchronized at the G2-phase when they would generally be delayed at the G2/M-phase [47,48,49]. CapHV-1, which infects peripheral blood mononuclear cells (PBMCs), can arrest cells at the G0/G1-phase, and CapHV-1 (caprine herpesvirus-1) significantly induces apoptosis during this stage [50]. As noted in research on Marek’s disease virus (MDV), which is from the same genus as DPV, chicken embryo fibroblasts (CEFs) undergo S-phase cell cycle arrest when infected with MDV, and MDV-encoded tegument VP22 plays an important role in the blocking process, which is consistent with the results of this study with DPV [51]. In summary, we further showed that α-herpesviruses can dynamically cause cell cycle arrest, although the mechanism of this activity may vary among different types of cells.

The results of this study showed that DPV causes S-phase cell cycle arrest and increases intracellular ROS levels, the latter of which decreases the MMP and induces apoptosis. Furthermore, caspase-3, caspase-7, caspase-8, and caspase-9 are involved in DPV-induced apoptosis, suggesting that DPV depends on the caspase protein family to induce apoptosis.

## Figures and Tables

**Figure 1 viruses-11-00196-f001:**
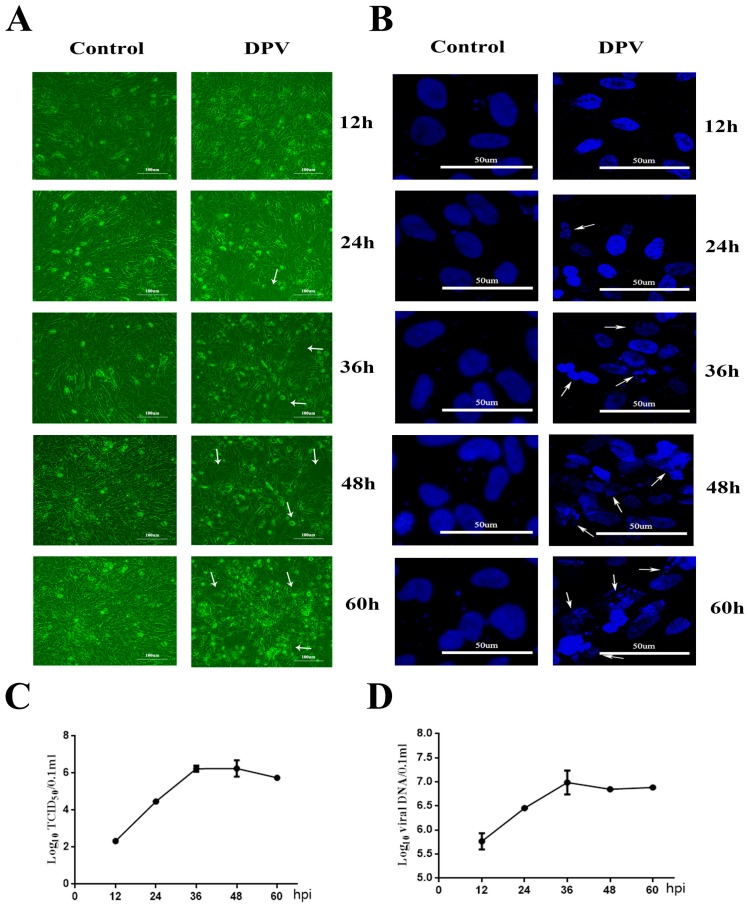
Cytopathic effects (CPEs) induced by duck plague virus (DPV) in duck embryo fibroblasts (DEFs). (**A**) Cellular morphological changes in cells infected with DPV for the indicated number of hours. At 24, 36, 48, and 60 hpi (hours postinfection), the arrows indicate that infected cells appeared to have cellular fragmentation and plaques. (**B**) Nuclear morphological changes in cells infected with DPV for the indicated number of hours. At 24, 36, 48, and 60 hpi, the arrows indicate that nuclei of infected cells appear appeared as fragmented and marginated typical apoptotic bodies. (**C**) Viral titers were determined at the indicated time points by measuring the TCID_50_ for the DEFs. All titrations were carried out in three independent experiments. The titers obtained were averaged, and the standard error of the mean was calculated for each time point. (**D**) Quantitative analysis of viral DNA by quantitative real-time PCR assay. Viral DNA detection was carried out in three independent experiments. The titers obtained were averaged, and the standard error of the mean was calculated for each time point.

**Figure 2 viruses-11-00196-f002:**
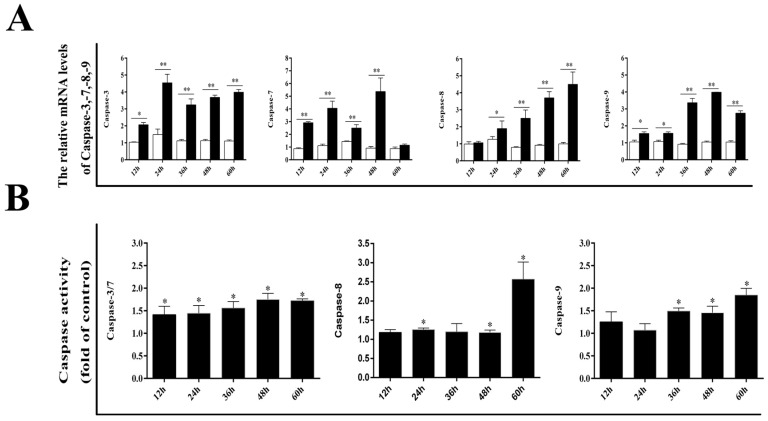
Effects of DPV infection on the caspase family. (**A**) mRNA expression levels of caspase-3, caspase-7, caspase-8, and caspase-9. (**B**) Activities of caspase-3, caspase-7, caspase-8, and caspase-9. The data are presented as the means ± SD of three independent experiments. * *p* < 0.05 and ** *p* < 0.01, compared with the control group.

**Figure 3 viruses-11-00196-f003:**
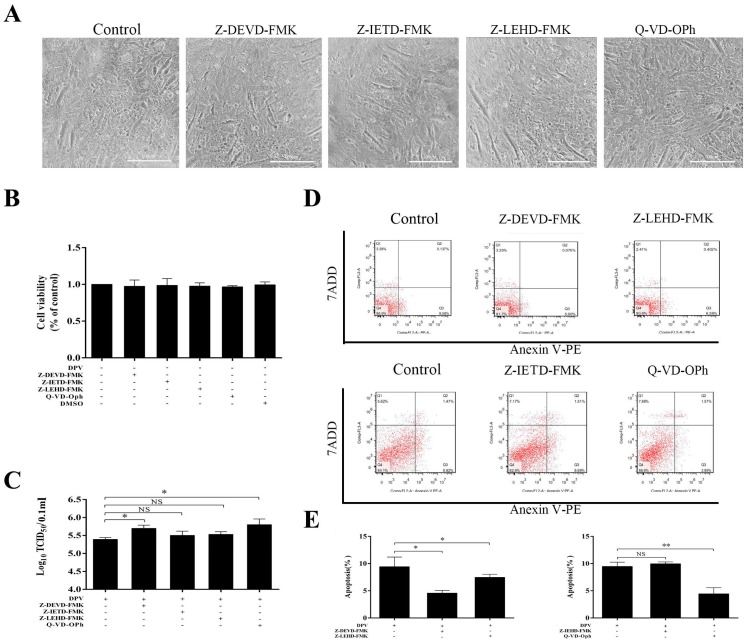
Z-IEHD-FMK, Z-DEVD-FMK, Z-IETD-FMK, and QVD-P-Oh inhibit apoptosis induced by DPV. (**A**) Cellular morphological changes following treatment with Z-IEHD-FMK, Z-DEVD-FMK, Z-IETD-FMK, and QVD-P-Oh for 2 h and detected at 48 h. (**B**) Changes in DEF viability following treatment with Z-IEHD-FMK, Z-DEVD-FMK, Z-IETD-FMK, and QVD-P-Oh for 2 h were measured using an MTT assay kit at 48 h. (**C**) DEF cells were pretreated with inhibitors for 2 h and then infected with DPV for 48 h. After incubation, the viruses were collected, and the viral titers were determined and presented as log10 TCID_50_/mL. (**D**) DEFs were pretreated with Z-IEHD-FMK or Z-DEVD-FMK for 2 h and then infected with DPV for 36 h. DEFs were pretreated with Z-IETD-FMK or QVD-P-Oh for 2 h and then infected with DPV for 48 h. Apoptosis was detected by FCM. (**E**) Histogram of the percentage of apoptotic cells; DEFs were pretreated with Z-IEHD-FMK or Z-DEVD-FMK for 2 h and then infected with DPV for 36 h. DEFs were pretreated with Z-IETD-FMK or QVD-P-Oh for 2 h and then infected with DPV for 48 h. The data are presented as the means ± SD of three independent experiments. * *p* < 0.05 and ** *p* < 0.01, compared with the control group.

**Figure 4 viruses-11-00196-f004:**
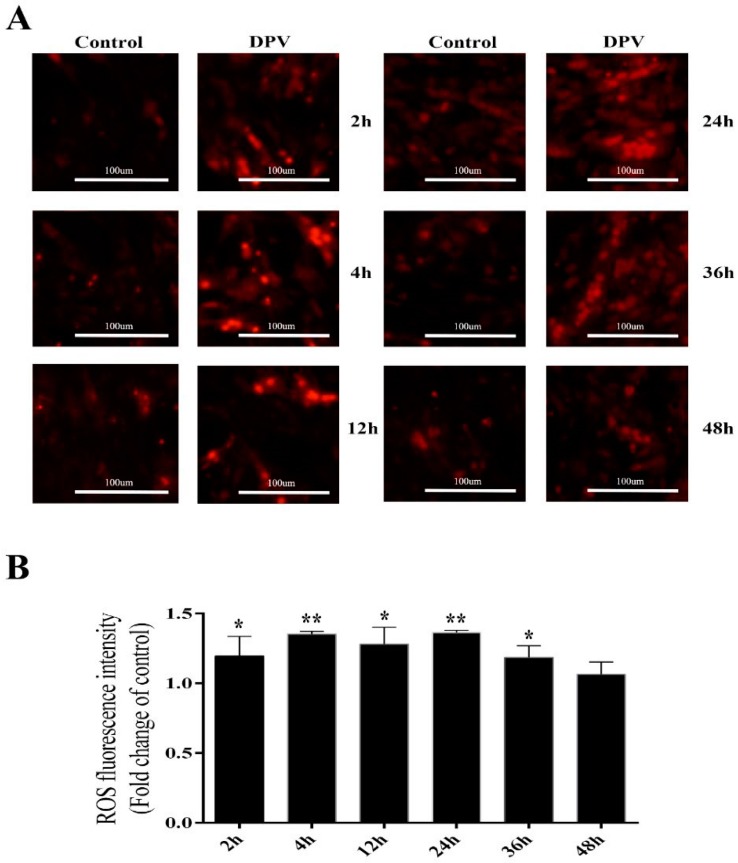
Determination of ROS levels in DEFs. (**A**) Assessment of ROS levels in DEFs using an intracellular ROS detection kit and fluorescence microscopy; the red color indicates intracellular ROS. (**B**) Assessment of ROS levels in DEFs using an intracellular ROS detection kit and a multifunctional microplate reader. The data are presented as the means ± SD of three independent experiments. * *p* < 0.05 and ** *p* < 0.01, compared with the control group.

**Figure 5 viruses-11-00196-f005:**
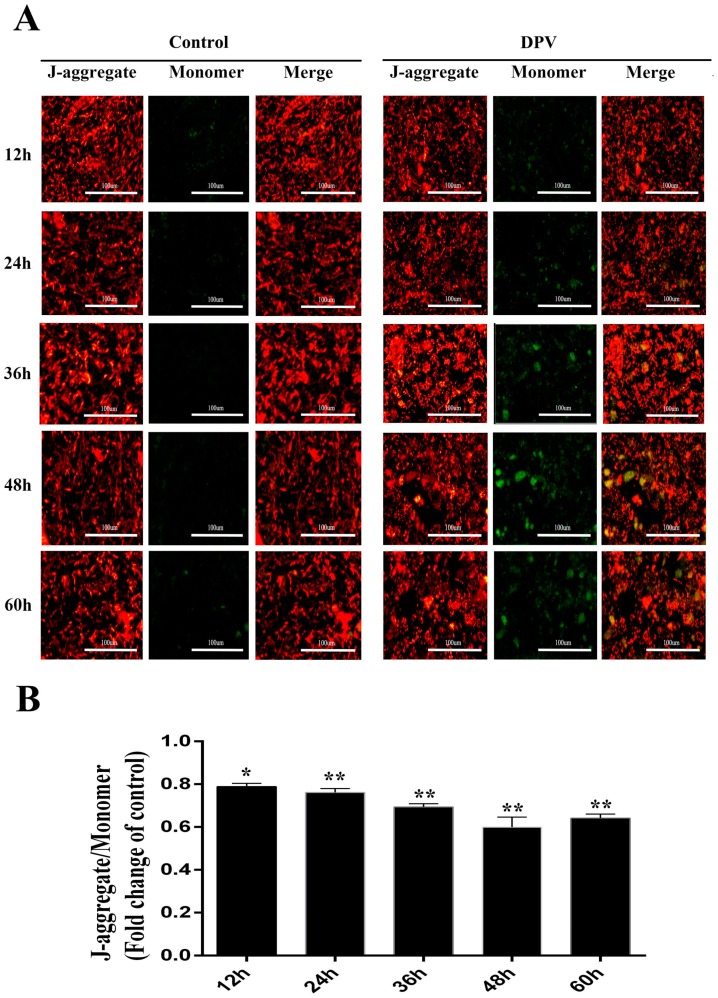
Mitochondrial membrane potential (MMP) determination in DEFs. (**A**) Assessment of DEF MMP through JC-1 staining and fluorescence microscopy. Mitochondria with normal membrane potential are indicated in red, and mitochondria with reduced membrane potential are indicated in green. (**B**) Assessment of DEF MMP though JC-1 staining and a multifunctional microplate reader. The data are presented as the means ± SD of three independent experiments. * *p* < 0.05 and ** *p* < 0.01, compared with the control group.

**Figure 6 viruses-11-00196-f006:**
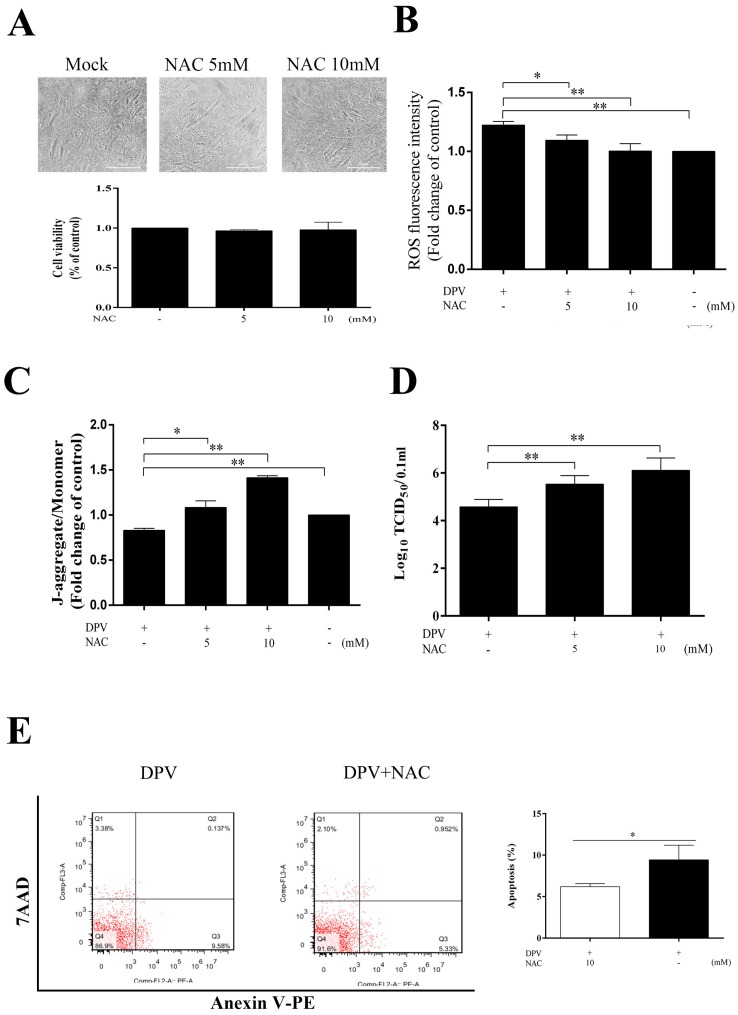
The ROS scavenger NAC reduces ROS levels, increases the MMP, and inhibits apoptosis. (**A**) Changes in cell viability following DEF pretreatment with 5 or 10 mM NAC for 2 h and detected by MTT assay at 36 h. (**B**) Pretreatment with 5 or 10 mM NAC decreased ROS in DPV-infected cells; uninfected cells were used as a control. (**C**) NAC (5 or 10 mM) increased the MMP in DPV-infected cells; uninfected cells were used as a control. (**D**) NAC (5 or 10 mM) promoted viral replication. (**E**) NAC (10 mM) inhibited apoptosis induced by DPV. The data are presented as the means ± SD of three independent experiments. * *p* < 0.05 and ** *p* < 0.01, compared with the control group.

**Figure 7 viruses-11-00196-f007:**
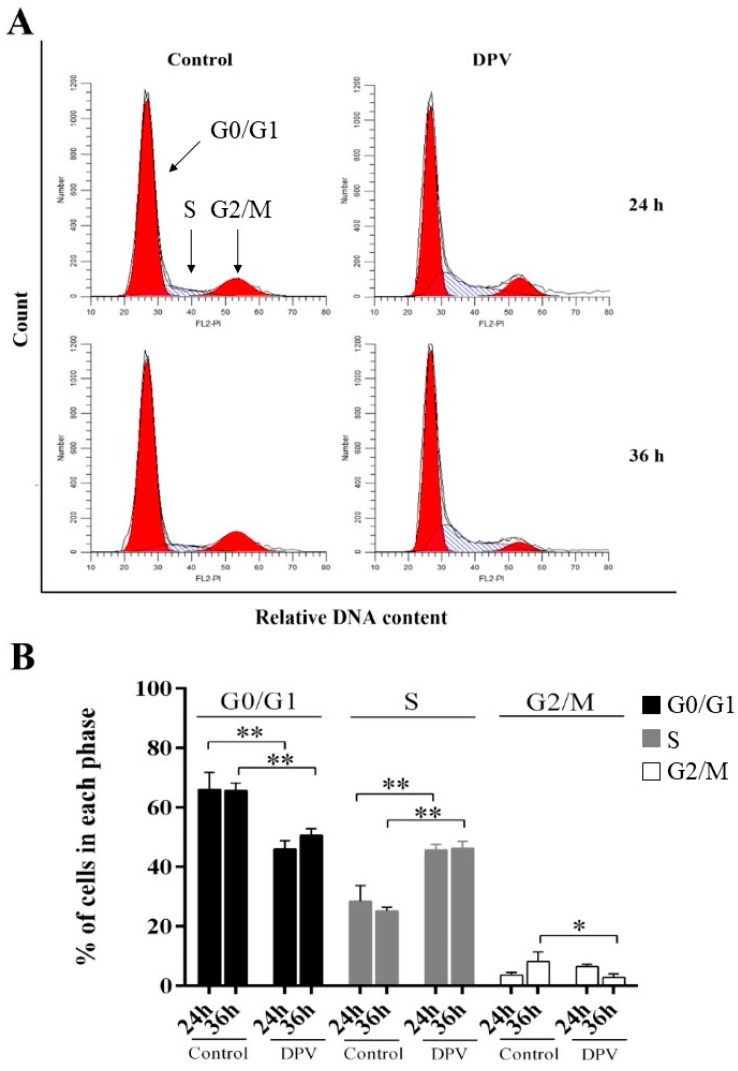
DPV induces cell cycle arrest at the S-phase. (**A**) DEFs were mock-infected (control) or infected with DPV. Cells were collected postinfection (p.i.) at the times indicated for FACS analysis of the cell cycle. (**B**) Histogram of the percentage of DEFs in the G0/G1-, S-, and G2/M-phases of the cell cycle, with the percentage of cells in each phase of the cell cycle shown. The data are presented as the means ± SD of three independent experiments. * *p* < 0.05 and ** *p* < 0.01, compared with the control group.

**Table 1 viruses-11-00196-t001:** Primers for qRT-PCR analysis of gene expression.

Target Gene		Primer Sequences	Gene Accession Number
**Caspase-3**	For	5′ TGGTGTTGAGGCAGACAGTGGA 3′	XM_005030494
	Rev	5′ CATTCCGCCAGGAGTAATAGCC 3′	
**Caspase-8**	For	5′ GGTGATGCTCGTCAGAAAGGTG 3′	XM_013094737
	Rev	5′ AGCCATGCCCAAGAGGAAGT 3′	
**Caspase-9**	For	5′GCTGCTTCAACTTCCTCCGTAA 3′	XM_013095294
	Rev	5′ CATCTCCACGGACAGACAAAGG 3′	
**β-Actin**	For	5′ CCGGGCATCGCTGACA 3′	NM_001310421
	Rev	5′ GGATTCATCATACTCCTGCTTTGCT 3′

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
