# Peer review of "Duck Plague Virus Promotes DEF Cell Apoptosis by Activating Caspases, Increasing Intracellular ROS Levels and Inducing Cell Cycle S-Phase Arrest"

_viruses, 2019, doi:10.3390/v11020196_

Round 1
Reviewer 1 Report
Overall I find this study of interest and was carried out carefully. The information of how Duck Plague virus induces apoptosis should be useful for future understanding as to how to control this type of virus infection. While I generally agree with the overall conclusions of the manuscript, I have some specific concerns on the presentation of the work that involve 1) lack of detailed figure legends 2) lack of some needed controls and 3) lack of proper labeling of the figures. There are some english language issues but they are fairly minor.
Concerns:
In figure 1: they show arrows in the figure 1A and 1B but nowhere in the legend or the manuscript do they describe what the arrows are pointing at exactly, this needs to be added to the manuscript. Also, in Figure 1 they indicate p<0.05 and 0.01 but there are no * in Figure 1B. I am not sure if they want to refer to statistics in this figure but if they do they need to add the *'s.
In Figure 2: From my understanding, caspases are proenzymes which become activated to active forms during triggering of apoptosis. The fact that the mRNA for these enzymes go up does not necessarily mean the active enzymes themselves go up. In fact in their activity assay there is only a 1.5 fold increase in caspase activity which I find very low assuming that the infected cells are all infected with virus. Thus, I am not sure one can conlcude that caspase activation is the cause of apoptosis. To improve this assessment I think the authors should do western blots for PARP cleavage and also look for the active form of the enzymes on the blots. This would be more convincing that the small increases in activity they see actually lead to a functional change in the cells.
In Figure 3: They show viability data for cells treated with caspase inhibitors but they do not say what cells or how long they were treated nor do they indicate what the concentrations of the inhibitors are. This must be added.
In Figure 4: In A they really should indicate in the legend what the red color is depicting. Also, in Figure 4B there are (again) no *'s showing what is or is not significant yet they show info for statistics in the legend. I like the clear increase in red color meaning increases in ROS but it does not show up that well in Figure B which seems odd to me.
In Figure 5: Figure 5A is very nice. But figure 5B again has no *'s for significance and this constant lack of attention to detail is concerning.
In Figure 6: Again the control treatments with NAC to show no killing is good but you do not say how long the cells were treated. Also, Figure 6B should also show uninfected cells as controls to see if one would see this drop even in uninfected cells.
In Figure 7. The histograms should be labeled in a way that points to the different phases of the cell cycle. There is hashed grey area but nothing in the legend describes what it is.
Note in the materials and methods section there are numerous pieces of info lacking. For example no indication of what the caspase inhibitors were dissolved in and if a control was used with the same solution and they do not indicate the caspase-inhibitor concentrations.
Other minor issues:
line 31: need the word "of" in understanding (of) the pathogenesis
line 42-43: not sure what they mean by "cell concentration" but maybe they mean "condensation"?
line 67: need the word "and" in DPV pathogenesis (and) the apoptotic pathways....
line 120: error here please check "a and"?
line 192: need "48" in control cells at (48) hpi (Figure 4B)
line 270: remove the word "with" in "increased with over time..."
line 274: should be share not shares
line 287: add the word "it" in lower MMP, (it) is surprising
Author Response
Dear Editors and Reviewers:
Thank you for your letter and for the reviewers’ comments concerning the manuscript that we submitted to Viruses. The reviewers’ comments are all valuable and were very helpful for revising and improving our paper and guiding our research. Based on the comments and suggestions, we have made careful modifications to the original manuscript. We hope the new manuscript will meet your journal’s standards. Below, you will find our point-by-point responses to the reviewers’ comments and Editorial Board Member comments.

Reviewer 2 Report
The manuscript by Zho et al., describes studies that show Duck Plaque Virus induces apoptosis in duck embryo fibroblast cells through both intrinsic and extrinsic pathways. In addition, DPV infection causes S-phase arrest in infected cells where intercellular ROS increases and mitochondrial membrane potential decrease. These finding increase our knowledge of apoptosis induction by DVP, but are not hypothesis-driven, do not overturn current thought or present novel experimental procedures. This work is internally consistent, but incremental in nature and does not provide future goals. That said, the virus is novel and under-studied, the data is solid and not over-interpreted. The following comments are presented to improve clarity and address concerns.
1. It is very difficult to claim one’s work is the first to show/prove something. Please remove these references.
2. Please check your statistics. TCID50 determined on log-dilutes is a technique which should be analyzed by non-parametric statistics, unless the authors can cite published information to the otherwise.
3. Check grammar in line 120.
4. In Fig. 1 A & B, it is unclear what the arrows indicate. Also what is the stain used to generate the green cells in the figure? Figure 1C is confusing. The y-axis suggests genome copies were determined, but the text indicates that RT-qPCR was used. Was DPV DNA or mRNA quantified; if mRNA, what virus genes were investigated? Also, how does one correlate an approximate 4-log increase in virus genomes and only an approximate 1 log increase in infectious virus?
5. In Fig. 2A, was delta-delta Ct analysis used? If not, how was the assay normalized to cell number?
6. Fig. 3 should be modified. The panels lettering is confusing with two A’s and 3B’s.
7. Figure 4a does not show the significance indicated in line 191. Also Fig. 5B does not show the significance indicate in line 207.
8. Is something missing in line 192?
9. Is there a way of showing the results presented in Fig. 6 reflect infected cells or a bystander effect?
10. Fig. 7 legend needs to be expanded to explain the different traces in panel A.
11. Please spell out Caprine Herpesvirus-1 in line 305.
Author Response

(The authors gave the same response as above.)

Round 2
Reviewer 1 Report
I am happy with the changes that were made to improve the manuscript and it is acceptable for publication. However, in many of the edited new portions the english grammar is incorrect. Here I will state some of them but I hope the people that put the proofs together can make these corrections throughout the manuscript.
Also, Line 86 says "wrote" should be change to "described"
line 29 says "prove" should be changed to shows
Line 98 says "were" should say "was"
Line 158 should say "arrows indicate that infected cells"
line 161 says "was" should be "is"
Line 164 should state "arrows indicate that nuclei of infected cells appear"
Line 174 should say "viral titers"
Line 176 should say "experiments"
these are some but these types of errors are throughout and most often in the figure legends, especially and ironically in the corrected ones
Author Response
Dear Editors and Reviewers:
Thank you for your letter and for the reviewers’ comments concerning the manuscript that we submitted to Viruses. The reviewers’ comments are all valuable and were very helpful for revising and improving our paper and guiding our research. Based on the comments and suggestions, we have made careful modifications to the manuscript. We hope the new manuscript will meet your journal’s standards.
